# Effect of Microencapsulation Techniques on the Stress Resistance and Biological Activity of Bovine Lactoferricin-Lactoferrampin-Encoding *Lactobacillus reuteri*

**DOI:** 10.3390/foods11203169

**Published:** 2022-10-11

**Authors:** Xueying Wang, Weichun Xie, Senhao Zhang, Yilan Shao, Jiyao Cai, Limeng Cai, Xiaona Wang, Zhifu Shan, Han Zhou, Jiaxuan Li, Wen Cui, Li Wang, Xinyuan Qiao, Yijing Li, Yanping Jiang, Lijie Tang

**Affiliations:** 1College of Veterinary Medicine, Northeast Agricultural University, Harbin 150030, China; 2Heilongjiang Key Laboratory for Animal Disease Control and Pharmaceutical Development, Northeast Agricultural University, Harbin 150030, China

**Keywords:** lactoferricin-lactoferrampin (LFCA), *Lactobacillus reuteri*, microcapsules, spray dry

## Abstract

Bovine lactoferricin-lactoferrampin-encoding *Lactobacillus reuteri* (LR-LFCA) has been found to benefit its host by strengthening its intestinal barrier. However, several questions remain open concerning genetically engineered strains maintaining long-term biological activity at room temperature. In addition, probiotics are vulnerable to harsh conditions in the gut, such as acidity and alkalinity, and bile salts. Microencapsulation is a technique to entrap probiotic bacteria into gastro-resistant polymers to carry them directly to the intestine. We selected nine kinds of wall material combinations to encapsulate LR-LFCA by spray drying microencapsulation. The storage stability, microstructural morphology, biological activity, and simulated digestion in vivo or in vitro of the microencapsulated LR-LFCA were further evaluated. The results showed that LR-LFCA had the highest survival rate when microcapsules were prepared using a wall material mixture (skim milk, sodium glutamate, polyvinylpyrrolidone, maltodextrin, and gelatin). Microencapsulated LR-LFCA increased the stress resistance capacity and colonization abilities. In the present study, we have identified a suitable wall material formulation for spray-dried microencapsulation of genetically engineered probiotic products, which would facilitate their storage and transport.

## 1. Introduction

The intestinal mucosa is the first site for the dynamic interaction of enteric pathogens with the host. Probiotics are live commensal microbes that have been proven to be effective against intestinal pathogens and bring health benefits when consumed in adequate quantities [1]. There are a number of formulas and dietary products that provide infants with probiotics (lactic acid bacteria) that are intended to positively affect gut health [2]. Probiotics have been shown to modulate the immune response, improve epithelial and mucosal barrier function, inhibit pathogenic bacteria, and alter the gut microbiota composition in an inflamed gut [3]. To exert their desired effects, the nature of the probiotic itself needs to be considered. Thus, one can design next-generation bioengineered probiotic strains to incorporate desirable traits.

Antimicrobial peptides (AMP) are amphiphilic, consisting of both cationic and hydrophobic components, and can electrostatically bind to anionic bacterial membranes or other anionic targets [4]. It is thought that AMP plays an important role in the immune system of a wide range of organisms, presenting the first line of defense against pathogens of diverse types [5]. The Bovine lactoferrin is approximately 70% homologous to the Human lactoferrin but has stronger antimicrobial properties. Microbiocidal properties of lactoferrin include cell membrane disruption, iron sequestration, inhibition of adhesion of bacteria to the host cell, and prevention of biofilm formation [6]. However, the complicated synthesis and purification procedures for these molecules are expensive and laborious. Bovine lactoferricin (Lfcin B) and lactoferrampin (Lfampin) are two antimicrobial peptides derived from pepsin-digested bovine lactoferrin [7]. The lactoferrin peptide enhances the host’s ability to defend against pathogens and could produce anti-tumor immunity [8]. It is known that *Lactobacillus reuteri* colonizes a variety of mammals, including humans, and improves their intestinal barrier by reducing microbial translocation from the gut lumen to the tissues [9]. As an internationally recognized safe probiotic, *Lactobacillus* or *Lactococcus* has been considered a suitable delivery vehicle to express the fusion of Lfcin B with Lfampin (LFCA) for regulating the intestinal mucosal immunity, improving intestinal microbiota diversity, and preserving the mucosal integrity [10,11,12].

Probiotics have long been used as gastrointestinal therapeutics, and they need to adapt to various environmental differences in the intestinal environment [13]. The oral delivery of probiotic species to the gut microbiome is of pronounced interest; however, the environmental complexity within the gastrointestinal tract decreases the oral bioavailability and limits the intestinal colonization of the administered probiotics. The viable microorganism number is a critical criterion for a successful probiotic product. According to the World Health Organization, adequate amounts of probiotics [10^6^–10^7^ colony-forming units (CFU)/g or mL] are prerequisites to confer health benefits [14,15]. Most probiotic cultures used in industrial applications are commercialized in dried form to reduce transportation costs and storage space [16]. Spray drying has been typically used in the chemical engineering, dairy, and pharmaceutical industries [17]. This technique involves the conversion of a continuous liquid film into droplets that meets a hot, dry airflow [18]. Energy consumption in spray drying is generally 10 times less than in freeze drying and can be carried out continuously. Moreover, spray-dried products can be stored at room temperature, reducing cold storage costs [19]. In the case of fixed equipment, the properties of the microcapsules depend on the nature of the feed flow and operating parameters such as flow rate and inlet temperature. This study mainly studies the effect of different embedding materials on the embedding effect of microcapsules [20,21]. Common protective agents for spray-dried products include monosaccharides, disaccharides, polysaccharides, amino acids, proteins, minerals, and their combinations [22]. Suitable protectants provide physical support for the encapsulated bacteria, resist the adverse environment of the gastrointestinal tract, and maintain cell viability and original biological properties of the bacteria during processing and storage [23]. Therefore, the use of appropriate wall materials is a crucial consideration for whether the spray-dried products can play a probiotic role.

As a result of using different wall materials, microencapsulation can exhibit a wide range of physicochemical properties based on the structure and properties of the materials used [24]. The current study aimed to evaluate the effect of various wall materials on the bioactive-compound retention and stability in microencapsulated bovine lactoferricin-lactoferrampin-encoding *L. reuteri* (LR-LFCA). The pig is the best model for human intestinal biology and diet research due to the intestinal anatomy similarity and the presence of human intestinal microbes [25,26,27]. Thus, the pig model is an ideal choice for evaluating the colonization ability and the immunomodulatory effect of probiotic products in the human intestine.

In this study, we used spray drying to prepare LR-LFCA into different kinds of protective agent-coated probiotic products and explored their microstructure, survival rate, protein expression, and bacteriostatic activity. The ability of microencapsulated LR-LFCA (MC LR-LFCA) to survive in harsh environments and colonize the piglet intestine was validated using in vitro and in vivo experimental models. We hypothesized that the idealized wall material could enhance the stress resistance of recombinant probiotics expressing AMP and better colonize the gut without affecting their biological activity.

## 2. Materials and Methods

### 2.1. Bacterial Strains and Growth Conditions 

*L. reuteri* CO21 was isolated from the intestinal contents of pigs and obtained from our laboratory (Laboratory of Microbiology and Immunology, College of Veterinary Medicine, Northeast Agricultural University). The 16S rDNA sequence of this strain was deposited in GenBank (accession No. MK920155). LR-LFCA and *L. reuteri* CO21 transformed with a pPG612-EGFP plasmid (LR-CON) were constructed and preserved in our laboratory. Genetically modified strains of *L. reuteri* (MC LR-LFCA, LR-LFCA, and LR-CON) were grown in MRS broth (De Man, Rogosa and Sharpe, Oxoid, Hampshire, UK) at 37 °C for 16 h. *Staphylococcus aureus* CVCC546 and *Escherichia coli* CVCC10141 were cultured in LB broth (Oxoid, Hampshire, UK) for 12 h to reach 10^8^ CFU/mL.

### 2.2. Preparation of Microencapsulated LR-LFCA

LR-LFCA was cultured at 37 °C for 16–18 h, and the precipitates from the bacterial suspensions were collected via centrifugation at 3500× *g* for 5 min. Then, nine different wall materials were prepared (Table 1). After mixing the respective precipitates and wall materials, the final product was spray-dried. The spray dryer was operated at an inlet temperature of 140 °C and an outlet temperature of 60 °C. The wall materials with the highest survival rate were selected for subsequent in vitro biological activity and in vivo colonization research.

### 2.3. Polymerase Chain Reaction (PCR) and Western Blotting

The plasmid pPG-LFCA-E was extracted using a plasmid extraction kit (TIANGEN, Beijing, China). The sequences of PCR primers were PCR forward 5′-CACACTTCTGAAAAAAGGAGGGGAGACCACAACGGTTTCCCACTAGAAATAATTTT-3′ and PCR reverse 5′-CCGCCAAAACAGCCAGATCTGTTATCACTTGTACAGCTCGTCCATGCCGAGAGTG-3′. Protein samples of LR-LFCA, MC LR-LFCA, and LR-CON were prepared as follows: The cells were collected by centrifugation, treated with lysozyme (10 mg/mL), and disrupted by sonication (sonication parameters: 400 W; 20 min; 4  s on, 6 s off). Homogenate was supplemented with sodium dodecyl sulfate (SDS) solution (1%, PBS with), and the samples were boiled for 10 min. The protein samples were analyzed by Tricine-SDS-polyacrylamide gel electrophoresis (15%, *w/v*). Western blot assay was performed using Myc-tag antibody (1:1000) as the primary antibody (Affinity Biosciences, USA). A horseradish peroxidase (HRP)-conjugated goat anti-mouse IgG (1:5000) (Abcam, Cambridge, UK) was used as the secondary antibody.

### 2.4. Quantitative Analysis of the Expression of Recombinant LFCA-E Protein

An anti-bovine lactoferrin monoclonal antibody (prepared by our laboratory and diluted at 1:200) was used as the primary antibody for ELISA. The content of bovine lactoferrin peptide in the culture supernatants and cell lysates for each period (8−24 h after incubation, every two hours) was calculated and determined. To calibrate the plate, a standard curve was prepared using manually integrated bovine lactoferrin. The resulting standard curve was utilized to quantitatively analyze the recombinant protein expression.

### 2.5. In Vitro Antibacterial Activity Assays for Microencapsulated LR-LFCA 

The antibacterial activity of MC LR-LFCA in vitro was determined using the Oxford cup assay. Pathogenic bacteria (*S. aureus* CVCC546 and *E. coli* CVCC10141) were preserved and propagated in our laboratory. 200 μL suspension of these pathogenic bacteria was smeared onto LB agar plates after they had been cultivated in LB broth to an OD600 of 0.6–0.8. The supernatants of the LR-LFCA and MC LR-LFCA cultures were filtered (0.22 μm) to generate cell-free supernatants (CFS). Then, 200 μL of cell-free supernatants were added at the midpoint of the Oxford cup. After culturing the plate statically at 37 °C for 24 h, the diameter of the bacteriostatic ring was measured and recorded.

### 2.6. Biological Characterization of Microencapsulated LR-LFCA

LR-LFCA and MC LR-LFCA were cultured and grown at 37 °C to an OD_600_ of 1.0. Both recombinant *L. reuteri* were then inoculated (1%. *v*/*v*) in MRS broth and cultivated at 37 °C. The growth curves of the recombinant bacterial strains were examined by recording the increase in OD_600_ and the colony number visible to the naked eye on the MRS agar plate every 2 h for 24 h. Acid production curves for the two strains were also generated. The medium pH was detected with a pH electrode.

### 2.7. Survival Rates of LR-LFCA after Spray Drying

CFU counts before and after spray drying were compared to obtain the bacterial survival rate. Survival rate (%) S = (Nt × M1/N0 × V2) ×100%, where Nt is the biomass of the bacterial agent after spray drying (CFU/g), M1 is the total weight of the spray-dried powder (g), N0 is the biomass of bacterial suspension (CFU/mL), and V2 is the total volume of the bacterial suspension (mL).

### 2.8. Morphology of the Microencapsulated LR-LFCA 

Three groups of MC LR-LFCA with high survival rates were selected for analysis. The surface morphology of the microcapsules was analyzed with a field-emission scanning electron microscope (JEOL, Japan) at a voltage of 5 kV using a previously described method [28]. Images were obtained under different magnifications (10,000×, 5000×, and 2000×).

### 2.9. Resistance of Microencapsulated LR-LFCA to Acidic/Alkaline Conditions and Bile Salts

Microencapsulated and non-microencapsulated LR-LFCA were tested for their ability to tolerate high or low pH levels and the presence of porcine bile salts. LR-LFCA were incubated in modified MRS broth (pH, 2.0, 3.0 and 8.0; 0.2%, 0.3%, 0.4%, and 0.5% bile salts) at 37 °C for 3 h. They were then grown on MRS agar plates at 37 °C for 24 h until the colonies were apparent. The survival rate was determined by dividing the number of colonies visible in the different environments by the number of colonies visible in untreated MRS broth (without pH regulation).

### 2.10. Resistance of Microencapsulated LR-LFCA to Artificial Gastrointestinal Fluid 

The ability of LR-LFCA microencapsulated and non-microencapsulated to tolerate artificial gastrointestinal fluid was tested. LR-LFCA were placed in simulated gastric fluid (SGF, pH 2.0, pepsin) and incubated for 30 min, 60 min, 90 min, or 120 min at 37 °C. Besides, LR-LFCA were placed in simulated intestinal fluid (SIF, pH 7.5, trypsin) and incubated for 1 h, 2 h, 3 h, or 4 h at 37 °C. The survival rate was determined by dividing the number of colonies visible in the different environments by the number of colonies visible in untreated MRS broth.

### 2.11. Storage Stability of Microencapsulated LR-LFCA

Microencapsulated and non-microencapsulated LR-LFCA were stored separately in different environmental conditions (Samples were placed at −20 °C, 4 °C, 25 °C, 30 °C, and 37 °C, individually under vacuum and air, and the relative humidity was set to 0.33, 0.54, and 0.76.). The plate count method was used to determine the initial number of viable probiotics in the microencapsulated and non-microencapsulated LR-LFCA. The viable probiotic counts were detected every 7 days to determine different change tendencies under different situations.

### 2.12. Animals and Experimental Design

All experimental protocols were approved by the Animal Care and Use Committee of Northeast Agricultural University. The approval code was NEAUEC20220358. A corn-soybean meal-based basal diet was formulated to meet the NRC2012 (Nutrient requirements of swine, 2012) recommendations for the nutrient requirements. All the animal care and treatment methods complied with the standards described in the Laboratory Animal Management Regulations (revised 2016) of Heilongjiang Province, China.

Twelve 3-day-old piglets (Rong Chang pigs) were randomly allocated to 3 treatments within each litter and adjusted for an average body weight of 1.35 ± 0.25 kg (balanced gender). Three groups were given different treatments from day 4 to day 6 as follows: oral administration of (1) 5 mL PBS (CON, *n* = 4); (2) LR-LFCA (1.0 × 10^10^ CFU per piglet, dissolved in 5 mL of PBS, *n* = 4); (3) MC LR-LFCA (1.0 × 10^10^ CFU per piglet, dissolved in 5 mL of PBS, *n* = 4). Twelve piglets were sacrificed after anesthesia by i.m. injection with sodium pentobarbital (40 mg kg^−1^ BW).

### 2.13. Sample Collection

The sample collection scheme for MC LR-LFCA colonization in 3-day-old piglets was designed as follows: LR-LFCA or MC LR-LFCA was administered orally daily for 3 days in 3-day-old piglets. Intestinal mucosa samples of the duodenum, jejunum, ileum, cecum, and colon were collected from neonatal piglets. The intestinal mucosa was scraped with a scalpel, placed in sterile PBS, and homogenized. The homogenized samples were diluted serially and spread onto MRS agar plates containing 50 μg/mL chloramphenicol. The plates were incubated at 37 ℃ for 48 h, and the resulting colonies were counted to analyze the colonization ability of LR-LFCA in piglets. The remaining intestinal mucus was stored at −80 ℃. A comparative analysis of different intestines was performed using a species-specific PCR for LR-LFCA. The primers used to construct the standard plasmid (L-Reu) were L-Reu-F: 5′-GCGTTGATGTTGAAGGAATGAGCTTTG-3′ and L-Reu-R: 5′-CATCAGCAATGATTAAGAGAGCACGGCC-3′. A 10-fold dilution of L-Reu standard plasmids into DEPC-treated water was performed; 10^9^, 10^8^, 10^7^, 10^6^, 10^5^, 10^4^, and 10^3^ copies of each plasmid were used in a well plate. The plasmid copy number was correlated with CT using quantitative fluorescence PCR, and a standard curve was generated.

### 2.14. Statistical Analyses

Data were calculated and reported as the mean ± standard deviation (SD). The statistical significance of the differences was analyzed using a one-way analysis of variance, followed by multiple comparisons between groups using Tukey’s post hoc test. Differences with *p*-values less than 0.05 were considered significant and reported as * *p* ≤ 0.05 or ** *p* ≤ 0.01. All calculations were performed using the SPSS software (version 19.0, SPSS, Chicago, IL, USA).

## 3. Results

### 3.1. Preparation and Evaluation of Microencapsulated LR-LFCA

The MC LR-LFCA was highly stable, with a survival rate of 83.56% (Table 2; skim milk/sodium glutamate/polyvinylpyrrolidone/maltodextrin/gelatin). The results presented in Table 3 illustrate the survival of free and MC LR-LFCA in the presence of porcine bile salts. The MC LR-LFCA grew in porcine bile salts at concentrations ≤0.5% (*w*/*v*). Table 4 shows that, when incubated at 37 ℃, the MC LR-LFCA showed a high degree of survival in solutions of different pH (2.0, 3.0, 8.0). MC LR-LFCA also showed good tolerance to both simulated gastric fluid (SGF) and simulated intestinal fluid (SIF) (Table 5 and Table 6).

### 3.2. Stability of Microencapsulated LR-LFCA under Different Storage Conditions

In order to investigate the impact of different storage conditions on the MC LR-LFCA, we stored spray-dried products at −20, 4, 25, 30, or 37 °C for over 100 days, under two different storage atmospheres: air or vacuum (Figure 1 and Figure 2). The total number of cells contained in the microcapsules changed remarkably over time. The results showed that after 154 days of storing LR-LFCA at 4 °C in different conditions, the number of remaining live bacteria in the skim milk/sodium glutamate/polyvinylpyrrolidone/maltodextrin/gelatin microcapsules was 10^10^ CFU/g (Figure 1B). As humidity levels increased at relatively constant temperatures, survival rates declined. To investigate the storage stability of MC LR-LFCA under different environments, the number of viable cells in the microcapsules was analyzed after exposure to different humidity levels (approximately 0.33, 0.54, and 0.76; Figure 3A–C, respectively). We found that the most rapid rate of viable bacteria decline was observed at 25 °C and relative humidity of approximately 0.76 (Figure 3).

### 3.3. Morphology and Colonization of Microencapsulated LR-LFCA in Neonatal Piglets

Scanning electron microscopy (SEM) was used to observe the microstructure of the three different microcapsules with high survival rates (wall materials: SM:SG:PO:MD:GE, SM:SU:GR:GI, TH:SM:MD). We found that the microcapsule particles were not ruptured and showed irregular spherical structures, with pits and folds on the surface, maintaining structural integrity (Figure 4). This feature is common in microcapsules obtained using spray drying because of the rapid water evaporation during the process. As a result, the wall materials completely enclosed the bacterial cells, reducing their exposure to oxygen. After continuous oral administration of LR-LFCA or MC LR-LFCA (wall materials: SM:SG:PO:MD:GE) to neonatal piglets, we collected the intestinal mucus from individual intestinal segments and determined the presence of LR-LFCA colonization. The results of bacterial colony counting assays showed that LR-LFCA colonized the duodenum, jejunum, and ileum (Figure 5A–E). Gastrointestinal fluid allowed significantly lower LR-LFCA colonization levels, whereas those with encapsulation had significantly higher colonization levels. Their levels were higher in the group administered with encapsulated LR-LFCA than in the group administered with LR-LFCA alone (*p* < 0.01) (Figure 5A–E). Data from the real-time PCR assay showed the same trend as the bacterial colony counting results (Figure 5F–J). The copy numbers of *L. reuteri* genes in the group administered with encapsulated LR-LFCA were significantly higher than in the group administered with LR-LFCA. 

### 3.4. Biological Activity and Biological Functions of Microencapsulated LR-LFCA

In order to explore the biological activity of encapsulated recombinant bacteria, we selected the microcapsules with the highest survival rate (wall materials: SM:SG:PO:MD:GE). Plasmid isolation and PCR identification confirmed the presence of pPG-LFCA-E plasmid in MC LR-LFCA (Figure 6A). LFCA-E expression was detected by Western blot analysis (Figure 6B). Analyses of bacterial growth curves showed that MC LR-LFCA and LR-LFCA exhibited similar growth patterns (Figure 6C). The acid production capacity curve of MC LR-LFCA (Figure 6D) showed a linear increase during the log phase of growth. The rapid decrease in pH inhibited the growth of pathogenic bacteria. It is calculated between 8 h and 24 h whether LFCA-E protein expression has increased or decreased using ELISA. We found that the maximal LFCA-E protein expression of 1.41 μg/mg and 1.38 μg/mL was achieved at 18 h in LR-LFCA and MC LR-LFCA (Table 7). The antibacterial activity of MC LR-LFCA is presented in Table 8. The results showed different levels of growth inhibition against *S. aureus* CVCC546 and *E. coli* CVCC10141. Negative staining and electron microscopy revealed that LR-LFCA cell lysates damaged the morphology of *S. aureus* CVCC546 and *E. coli* CVCC10141, and LR-LFCA lysate treatment caused more membrane damage, protrusions, and filamentations in bacteria than control treatment (Figure 6E).

## 4. Discussion

This study presented an effective technique for encapsulating the probiotic *Lactobacillus reuteri* via spray drying. However, during drying, *L. reuteri* is prone to thermal inactivation. In order to avoid this damage, adding wall materials for encapsulation is considered one of the most effective treatments [29]. Wall materials may act as a barrier against external pressures at the *Lactobacillus* interface, inhibiting the interaction between *L. reuteri* and other ingredients. A reduction in the response of *L. reuteri* to changes in the external environment allows their sustained release under specific conditions [30]. Wall materials include proteins, lipids, carbohydrates, and wax. It has been shown that trehalose, monosodium glutamate, polyvinylpyrrolidone, and sucrose act as crucial membrane-protecting agents for cells during environmental stress conditions like heat treatment and dehydration [22,31]. Comparatively to non-encapsulated cells, *Lactobacillus* cells encapsulated in soy protein isolate, glucose, fructan, or sodium caseinate had higher stability under storage or simulated gastrointestinal conditions [32,33,34]. Gelatin is a good choice as a wall material. Aside from its good film-forming ability and water solubility, gelatin shows excellent emulsification, low cost, and is non-toxic [35]. Skim milk proteins protect cell membranes from the damage caused by high temperatures. Calcium ions can cause the aggregation of milk proteins that protect the cells. Therefore, skim milk is the most intensively studied wall material for encapsulating bacteria [36]. Maltodextrin is popular in food processing as it is economic, nutrient-rich, and provides the core material a formidable barrier to external damage [37]. 

Mixing various proteins and carbohydrates into composite wall materials creates a synergistic effect. Adding hydrocolloids to protein wall materials can enhance their surface activity and viscosity [38]. Moreover, small-molecule sugars can effectively fill the pores on the surface of microcapsules during their formation [39]. Spray-drying the composite wall material results in smooth and uniform surfaces, and particle agglomeration is greatly diminished in the microcapsules. We compared the entrapment efficiency of different protective agents, and all formulations demonstrated excellent entrapment efficiency. Among them, composite wall materials (skim milk, sodium glutamate, polyvinylpyrrolidone, maltodextrin, and gelatin) showed the best protection performance against bacterial cells, reaching a survival rate of 85.36%. It was combined with gelatin, maltodextrin, and skim milk to play a synergistic role in protecting bacteria. In addition, the amino group of sodium glutamate reacted with the carboxyl group of proteins to stabilize their structure and polyvinylpyrrolidone regulated thermal stability. In addition, all microencapsulation formulations kept the original physical characteristics of recombinant LR-LFCA. This suggests that wall materials have an excellent protective effect on bacteria.

Probiotics have been widely used in livestock and poultry; however, the storage and transportation of probiotics can be affected by external environments, which results in the reduction or even loss of activity [17]. Techniques such as microencapsulation have been applied to protect probiotics, aiming to enhance storage stability and all aspects of tolerance capacity [18]. Most aerobic bacteria cannot survive under a vacuum, and the shelf-life of food products is extended under these conditions. We store microcapsules in a vacuumed aluminum foil bag at 4 °C to protect microcapsules from degradation and oxidation caused by the external environment, such as light, heat, oxygen, etc., during the storage process. It can prevent the interaction of the components and achieve the effect of controlling the release of core materials from the microcapsules at a specific time point. Microcapsules with vacuum packaging reduced the changes in microbiological and physical properties that occur in the probiotics during storage [40,41]. LR-LFCA was also effectively isolated from oxygen by the microcapsules formed by the composite wall material. Three types of microencapsulated *Lactobacillus rhamnosus* GG (electrospraying, freeze drying, and spray drying) were selected for this investigation [42], and the results demonstrated that freeze-drying and spray-drying microencapsulation adequately protected *Lactobacillus*. As reported by Bustamante et al. [43], the survival rate of *Bifidobacterium* and *Lactobacillus plantarum* encapsulated with composite carriers prepared using spray drying exceeded 98% during storage. This study utilizes composite wall materials to encase LR-LFCA. Anions and cations of different wall materials form a highly stable polyelectrolyte complex when they aggregate into multilayer membrane structures [44]. Compared with the unencapsulated groups, MC LR-LFCA had increased stress tolerance against bile salts, gastrointestinal fluids, and different pH levels. Here, we highlighted that microencapsulation could protect probiotics against the gastrointestinal environment. This suggested that the MC LR-LFCA used in this experiment had a strong stress tolerance and persisted longer in vivo.

Significantly decreased *Lactobacillus* counts were mainly due to the oxidation of membrane lipids. Thus, humidity and temperature during storage are important factors affecting *Lactobacillus* activity and stability [45]. As reported by Ying et al. [46], spray-dried microcapsules conditioned at 4 °C, and 32% relative humidity showed higher performance during storage (4 °C, 25 °C; 32%, 57%, and 70% relative humidity). Zokti et al. [47] found that the half-life of catechin extracts encapsulated at 4 °C was 231−288 weeks, and their stability was much higher than those stored at 40 °C and 25 °C. In our experiments, microcapsules were stored in different environments (−20 °C, 4 °C, 25 °C, 30 °C, and 37 °C). Due to the differences in the preparation period of spray-dried microencapsulation, both start and end time points within the detection time of the storage period were not exactly the same. We found that the counts of LR-LFCA stored at temperatures of −20 °C, 4 °C, and 25 °C for 140 days and temperatures of 30 °C and 37 °C for 60 days were maintained above 10^10^ CFU/g, which could still exert a long-term probiotic effect. This suggests that wall materials isolate the core material from the external environment and reduce the influence of external factors, which prolongs LR-LFCA survival.

Spray drying microencapsulation not only overcomes long-term shelf-life issues but also ensures the stability of embedding bioactive components. Our study showed that the growth and acid production trends of MC LR-LFCA were similar to those of LR-LFCA, and spray-drying microencapsulation did not affect its protein expression and bacteriostatic activity. Microencapsulation technology has been reported to enhance the stability and bioavailability of food or probiotics [48,49]. However, the effect of microencapsulation on the biological activity of engineered strains transformed with plasmids has rarely been reported. Appropriate wall materials ensured the stability of recombinant plasmids in probiotic products during spray drying. Observation through transmission electron microscopy showed that the MC LR-LFCA could destroy the cell membrane structure of *Staphylococcus aureus* and *Escherichia coli*, which were similar to our previous research on LFCA [12], and further confirmed that the microencapsulation does not affect the recombinant protein production. This is an important basis for the actual production of engineered strains expressing foreign proteins.

Because of the anatomical, physiological, and immunological similarities of the human and porcine intestines, pigs have been considered an attractive model to study the mechanisms involved in intestinal diseases as well as the microbial interaction with the immune system [50,51]. Moreover, the immune system of pigs is over 80% similar to humans, and pigs have a similar susceptibility to pathogens as humans [52]. Furthermore, it has been previously shown that LFCA is expressed by recombinant *Lactococcus lactis subsp.* improved the growth performance and immunity of piglets and alleviated dextran sulfate sodium (DSS)-induced colonic injury in mice [9,10]. *L. reuteri* inhibits the growth of pathogenic bacteria without harmful effects on the body and localized colonization in the GI mucosa [53,54]. In terms of protecting probiotics during gastrointestinal transit, encapsulation offers significant advantages. A high number of viable cells can colonize the intestinal mucosa after microencapsulated probiotics resist stomach acid, bile, and digestive enzymes during gastric transit. Encapsulated recombinant *Lactobacillus plantarum* expressing M cell homing peptide fused BmpB protein was completely released from alginate/chitosan/alginate microcapsules in a simulated small intestinal fluid within 12 h [55]. Junzhang Lin et al. [56] investigated alginate-chitosan-alginate (ACA) microcapsules entrapping live bacterial cells in vivo properties and found that microcapsules were stable in the rat gastrointestinal tract, which was attributed to the enhanced resistance of the ACA microcapsules to enzymatic digestion. This study is the first comprehensive application of microecological agents prepared using recombinant LR-LFCA on piglets. Here, we orally administered MC LR-LFCA to weaned piglets to examine whether they could colonize the intestine. MC LR-LFCA appeared to have a higher colonization capacity.

## 5. Conclusions

In conclusion, the strengths of this study include the in-depth analysis of microencapsulated LR-LFCA, which was prepared using a wall material mixture (skim milk, sodium glutamate, polyvinylpyrrolidone, maltodextrin, and gelatin), showing 85.36% survival. Microencapsulated LR-LFCA increased the stress resistance capacity and colonization abilities. The biological activity of the encapsulated genetically engineered bacteria was not affected. This study lays the foundation for future microencapsulation applications of genetically engineered probiotics.

## Figures and Tables

**Figure 1 foods-11-03169-f001:**
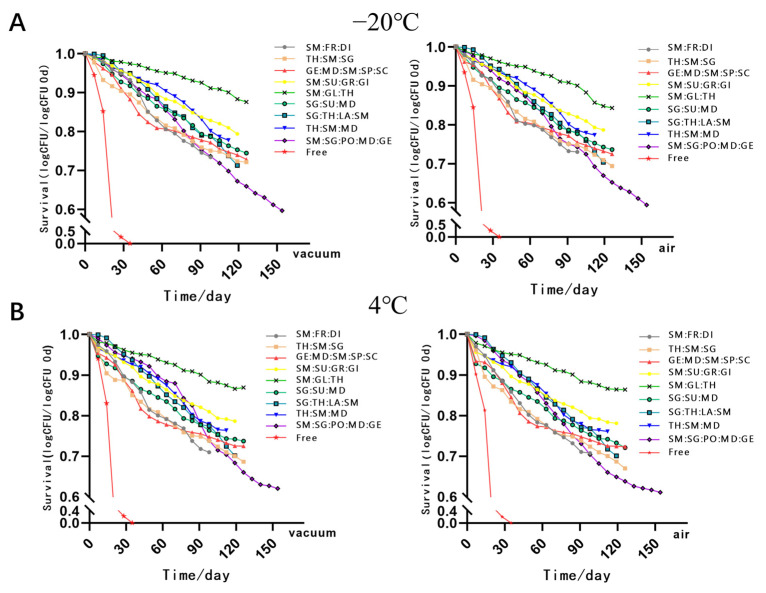
The number of viable cells in the microencapsulated LR-LFCA stored at −20 °C (**A**) and 4 °C (**B**) under two different storage atmospheres: air or vacuum.

**Figure 2 foods-11-03169-f002:**
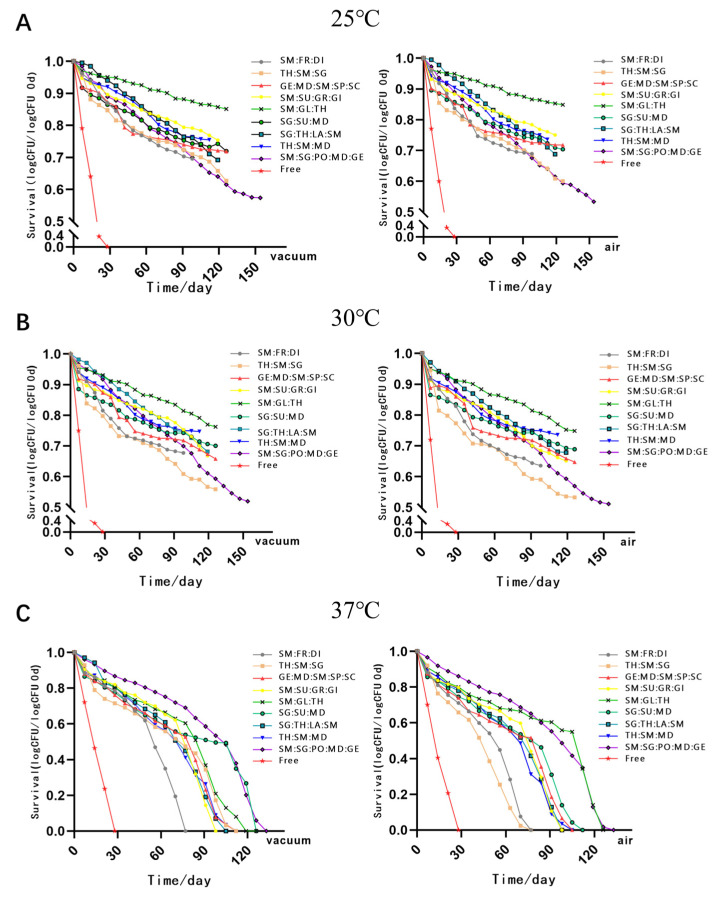
The number of viable cells in the microencapsulated LR-LFCA stored at 25 °C (**A**), 30 °C (**B**), and 37 °C (**C**) under two different storage atmospheres: air or vacuum.

**Figure 3 foods-11-03169-f003:**
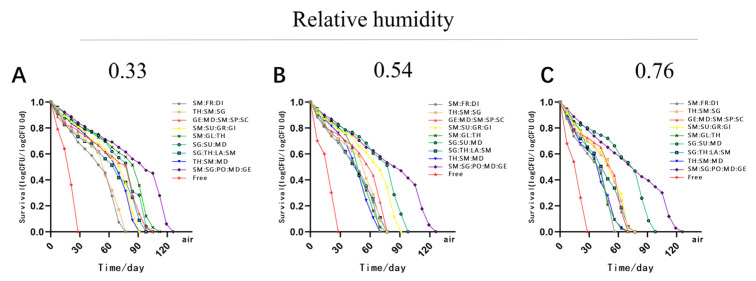
The number of viable cells in the microencapsulated LR-LFCA stored at 25 °C, with relative humidity values of approximately 0.33 (**A**), 0.54 (**B**), or 0.76 (**C**).

**Figure 4 foods-11-03169-f004:**
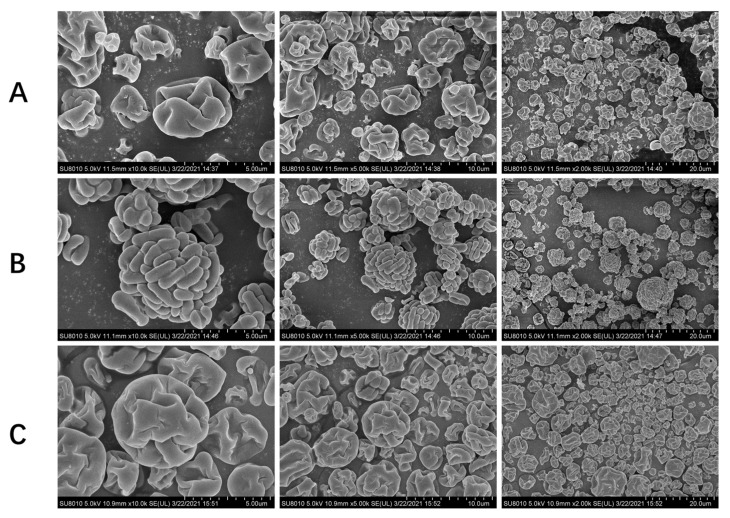
Scanning electron microscopy (SEM) images of microcapsules coated with different wall materials. (**A**) Skim milk/sodium glutamate/polyvinylpyrrolidone/maltodextrin/gelatin microcapsules. (**B**) Skim milk/sucrose/glycerol/glycine microcapsules. (**C**) Trehalose/skim milk/maltodextrin microcapsules.

**Figure 5 foods-11-03169-f005:**
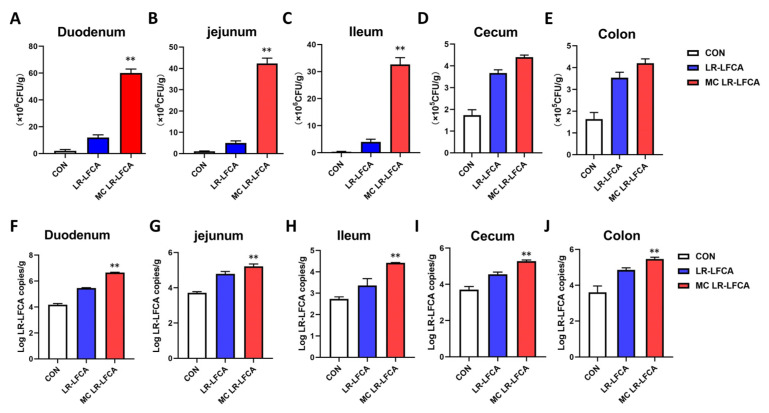
Microencapsulated LR-LFCA was assessed for its ability to colonize neonatal piglet intestinal mucosa. (**A**–**E**) The number of colonizing bacteria in the duodenum (**A**), jejunum (**B**), ileum (**C**), cecum (**D**), and colon (**E**) was verified using bacterial colony counting. (**F**–**J**) Quantification of total bacterial *L. reuteri* gene copy numbers in the duodenum (**F**), jejunum (**G**), ileum (**H**), cecum (**I**), and colon (**J**) were assessed using qPCR. Data are presented as the mean ± SD, ** *p* < 0.01 vs. LR-LFCA.

**Figure 6 foods-11-03169-f006:**
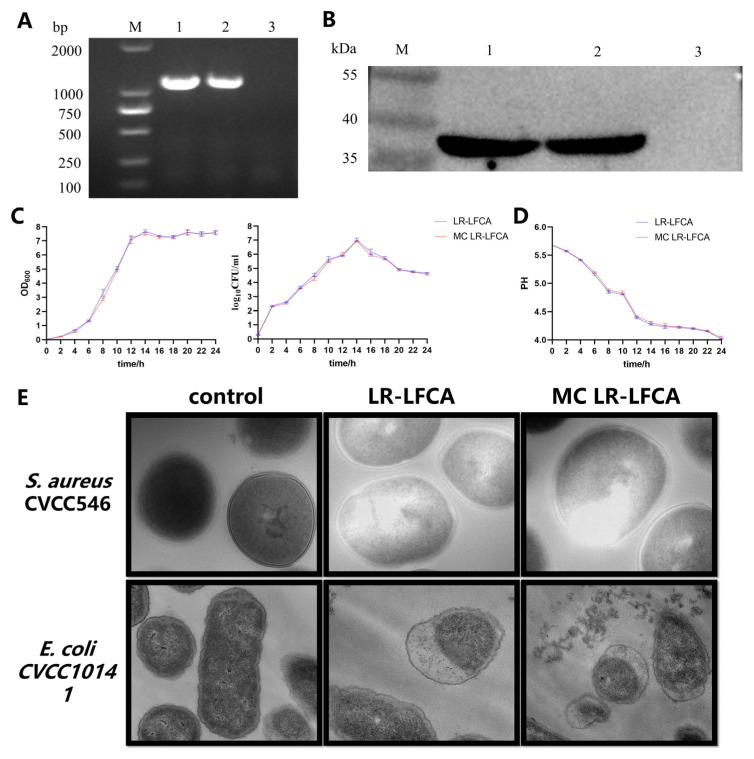
Protein expression, growth curves, and antimicrobial activity of microencapsulated LR-LFCA. (**A**) Identification of the recombinant plasmid pPG-LFCA-E and PCR products of approximately 1200 bp. 1: LR-LFCA. 2: MC LR-LFCA. 3: LR-CON. (**B**) LFCA-E protein expression was analyzed using western blotting, which resolved an immunoreactive band at 38 kDa. LR-LFCA, MC LR-LFCA, and LR-CON were cultured for 24 h, and the LFCA-E protein expression levels were assayed in the cell lysates. 1: The cell lysates of LR-LFCA. 2: The cell lysates of MC LR-LFCA. 3: The cell lysates of LR-CON. Irrelevant lanes were omitted. (**C**) Bacterial growth curve of LR-LFCA and MC LR-LFCA. OD values were determined at 600 nm using a microplate reader. (**D**) The pH changes in the culture medium of LR-LFCA and MC LR-LFCA. The pH changes in the culture medium were monitored every 2 h. The pH of the bacterial culture broth fell to 4.0 after 24 h of incubation. (**E**) Microstructural damage in bacteria treated with LR-LFCA or MC LR-LFCA lysates. Control, bacteria treated with PBS; LR-LFCA, bacteria treated with cell lysates from LR-LFCA; MC LR-LFCA, bacteria treated with cell lysates from MC LR-LFCA. Cells were analyzed by electron microscopy.

**Table 1 foods-11-03169-t001:** Composition of the wall materials used in this study.

Wall Materials (*w*/*w*)	Note
Free	Free
Skim milk 27%: Fructan 13.8%: Diatomite 13.8%	SM:FR:DI
Trehalose 10%: Skim milk 11%: Sodium glutamate 4%	TH:SM:SG
Gelatin 4%: Maltodextrin 10%: Skim milk 10%: Soy Protein lsolate 4%: Sodium caseinate 5%	GE:MD:SM:SP:SC
Skim milk 10%: Sucrose 5.73%: Glycerol 2.97%: Glycine 0.69%	SM:SU:GR:GI
Skim milk 35%: Glucose 15%: Trehalose15%	SM:GL:TH
Sodium glutamate 35%: Sucrose 50%: Maltodextrin 35%	SG:SU:MD
Sodium glutamate 2%: Trehalose 5%: Lactose 5%: Skim milk 10%	SG:TH:LA:SM
Trehalose 2.23%: Skim milk 2.5%: Maltodextrin 2.77%	TH:SM:MD
Skim milk 17.6%: Sodium glutamate 18.55%: Polyvinylpyrrolidone 25.5%: Maltodextrin 22.1%: Gelatin 14. 5%	SM:SG:PO:MD:GE

**Table 2 foods-11-03169-t002:** Survival of microencapsulated LR-LFCA in different wall materials after spray drying.

Wall Materials	Note	Survival (%)
Free	Free	0.00 ± 0.00
Skim milk 27%: Fructan 13.8%: Diatomite 13.8%	SM:FR:DI	40.67 ± 2.21
Trehalose 10%: Skim milk 11%: Sodium glutamate 4%	TH:SM:SG	15.99 ± 1.36
Gelatin 4%: Maltodextrin 10%: Skim milk 10%: Soy Protein lsolate 4%: Sodium caseinate 5%	GE:MD:SM:SP:SC	30.36 ± 3.57
Skim milk 10%: Sucrose 5.73%: Glycerol 2.97%: Glycine 0.69%	SM:SU:GR:GI	66.67 ± 5.49
Skim milk 35%: Glucose 15%: Trehalose15%	SM:GL:TH	43.81 ± 4.18
Sodium glutamate 35%: Sucrose 50%: Maltodextrin 35%	SG:SU:MD	56.38 ± 3.15
Sodium glutamate 2%: Trehalose 5%: Lactose 5%: Skim milk 10%	SG:TH:LA:SM	42.73 ± 7.26
Trehalose 2.23%: Skim milk 2.5%: Maltodextrin 2.77%	TH:SM:MD	64.17 ± 4.51
Skim milk 17.6%: Sodium glutamate 18.55%: Polyvinylpyrrolidone 25.5%: Maltodextrin 22.1%: Gelatin 14.5%	SM:SG:PO:MD:GE	83.56 ± 6.12

**Table 3 foods-11-03169-t003:** Tolerance of microencapsulated LR-LFCA to bile salts.

Samples	Survival (%)			
	0.2% Bile Salt	0.3% Bile Salt	0.4% Bile Salt	0.5% Bile Salt
Free	2.59 ± 0.35	1.92 ± 0.44	1.33 ± 0.37	0.96 ± 1.08
SM:FR:DI	70.86 ± 2.06	31.06 ± 0.76	26.26 ± 2.53	25.8 ± 0.90
TH:SM:SG	62.69 ± 2.55	31.84 ± 1.55	21.14 ± 1.15	14.15 ± 0.17
GE:MD:SM:SP:SC	61.15 ± 1.02	20.21 ± 1.28	9.7 ± 0.50	2.86 ± 0.03
SM:SU:GR:GI	49.50 ± 1.65	26.13 ± 1.11	12.25 ± 1.65	7.77 ± 1.15
SM:GL:TH	68.25 ± 1.63	43.41 ± 1.88	32.11 ± 3.04	16.65 ± 0.52
SG:SU:MD	59.83 ± 1.83	34.18 ± 1.62	16.01 ± 1.34	6.80 ± 0.24
SG:TH:LA:SM	64.67 ± 3.21	48.40 ± 1.28	13.74 ± 2.03	5.24 ± 0.90
TH:SM:MD	36.88 ± 1.57	14.63 ± 1.03	7.32 ± 0.79	2.45 ± 1.19
SM:SG:PO:MD:GE	49.97 ± 4.41	21.64 ± 2.30	17.64 ± 0.69	9.33 ± 0.69

**Table 4 foods-11-03169-t004:** Tolerance of microencapsulated LR-LFCA to acidic and alkaline solutions.

Samples	Survival (%)		
	pH2	pH3	pH8
Free	2.97 ± 0.22	5.30 ± 0.24	0.76 ± 0.19
SM:FR:DI	5.29 ± 0.05	22.40 ± 1.37	9.38 ± 0.86
TH:SM:SG	5.27 ± 0.27	11.15 ± 1.11	24.07 ± 0.82
GE:MD:SM:SP:SC	10.36 ± 0.63	29.42 ± 1.57	21.57 ± 1.57
SM:SU:GR:GI	13.33 ± 1.22	14.97 ± 1.37	22.60 ± 2.05
SM:GL:TH	14.93 ± 1.06	26.04 ± 1.95	23.56 ± 2.43
SG:SU:MD	14.81 ± 0.57	22.36 ± 2.89	31.63 ± 3.81
SG:TH:LA:SM	53.37 ± 1.62	73.02 ± 0.02	7.25 ± 0.75
TH:SM:MD	5.45 ± 0.04	16.07 ± 1.06	13.34 ± 0.90
SM:SG:PO:MD:GE	7.27 ± 0.22	11.57 ± 0.98	22.73 ± 2.15

**Table 5 foods-11-03169-t005:** Tolerance of microencapsulated LR-LFCA to simulated gastric juice.

Samples	Survival (%)			
	30 min	60 min	90 min	120 min
Free	83.60 ± 1.56	79.40 ± 5.51	72.17 ± 1.32	55.02 ± 3.14
SM:FR:DI	85.22 ± 1.22	80.55 ± 1.55	77.55 ± 1.41	70.55 ± 1.23
TH:SM:SG	87.90 ± 1.90	85.58 ± 0.58	80.63 ± 0.13	75.34 ± 0.34
GE:MD:SM:SP:SC	86.07 ± 0.07	83.47 ± 0.47	78.20 ± 1.20	70.76 ± 1.76
SM:SU:GR:GI	85.25 ± 2.25	80.03 ± 1.03	77.88 ± 1.88	70.18 ± 2.18
SM:GL:TH	85.02 ± 0.97	81.10 ± 0.89	79.52 ± 1.52	73.01 ± 1.01
SG:SU:MD	88.50 ± 1.50	82.25 ± 2.25	80.37 ± 0.37	74.17 ± 4.82
SG:TH:LA:SM	84.04 ± 0.04	81.35 ± 0.85	78.67 ± 0.32	70.17 ± 1.17
TH:SM:MD	85.11 ± 2.11	80.59 ± 3.59	75.48 ± 2.48	69.18 ± 1.18
SM:SG:PO:MD:GE	93.41 ± 0.52	89.97 ± 2.63	84.36 ± 0.92	77.32 ± 0.15

**Table 6 foods-11-03169-t006:** Tolerance of microencapsulated LR-LFCA to simulated intestinal juice.

Samples	Survival (%)			
	1 h	2 h	3 h	4 h
Free	62.60 ± 5.26	47.87 ± 1.81	40.94 ± 3.18	35.18 ± 1.32
SM:FR:DI	85.29 ± 1.29	80.00 ± 2.00	71.77 ± 1.77	63.11 ± 2.11
TH:SM:SG	81.20 ± 0.79	73.82 ± 1.82	70.25 ± 1.74	65.48 ± 1.48
GE:MD:SM:SP:SC	48.23 ± 1.76	45.95 ± 0.25	41.05 ± 1.07	37.15 ± 0.15
SM:SU:GR:GI	68.14 ± 1.14	53.51 ± 1.51	41.55 ± 1.44	35.44 ± 0.55
SM:GL:TH	85.42 ± 1.57	72.86 ± 0.13	59.94 ± 2.05	49.66 ± 1.33
SG:SU:MD	83.90 ± 1.90	74.71 ± 2.71	55.04 ± 2.04	46.19 ± 0.19
SG:TH:LA:SM	82.13 ± 0.86	73.52 ± 1.52	67.92 ± 1.92	55.37 ± 2.37
TH:SM:MD	84.12 ± 2.12	75.09 ± 0.09	45.15 ± 0.84	41.27 ± 0.72
SM:SG:PO:MD:GE	91.44 ± 2.27	85.58 ± 1.93	54.54 ± 0.40	40.65 ± 1.89

**Table 7 foods-11-03169-t007:** Expression levels of LFCA-E proteins in the cell lysates.

Incubation Time	LFCA-E Proteins in the Cell Lysates (μg/mg)	
	LR-LFCA	MC LR-LFCA
8 h	0.28 ± 0.05	0.32 ± 0.14
10 h	0.46 ± 0.08	0.43 ± 0.05
12 h	0.58 ± 0.10	0.51 ± 0.09
14 h	0.79 ± 0.12	0.70 ± 0.13
16 h	0.92 ± 0.21	0.95 ± 0.21
18 h	1.41 ± 0.30	1.38 ± 0.42
24 h	1.26 ± 0.11	1.20 ± 0.18

**Table 8 foods-11-03169-t008:** Antibiotic activity of the LR-LFCA against pathogenic bacteria.

The Tested Strains	The Size of the Bacteriostatic Circle Diameter (cm)		
	PBS	LR-LFCA	MC LR-LFCA
*S. aureus* CVCC546	0	0.63 ± 0.08	0.59 ± 0.06
*E. coli* CVCC10141	0	0.48 ± 0.07	0.43 ± 0.15

## Data Availability

Data is contained within the article.

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
