# Peer review of "Effect of Microencapsulation Techniques on the Stress Resistance and Biological Activity of Bovine Lactoferricin-Lactoferrampin-Encoding Lactobacillus reuteri"

_foods, 2022, doi:10.3390/foods11203169_

Round 1

Reviewer 1 Report

What seems particularly new to me about this study is the depth of it, such as the use of colonization studies and biological activity studies. I however have several questions:

The authors do not really explain why they used certain excipients. Also to me it seems that the different formulations were chosen in such a way that we cannot compare the influence of the different individual components on the results. This seems a bit unscientific to me. The authors also do not really explain why different formulations give different results.

In the studies related to the activity of the formulations, e.g. the animal studies, it is unclear to me if the used dosage was corrected for the survival after spray-drying. In other words: if the dosage given to the piglets or used in the activity studies was based on the cfu's before spray-drying then it is logical that the microencapsulated formulations, which give a higher survival, show much more colonization and activity. 

Why do the authors expect that the microencapsulated formulations should enhance colonization and the activity other then because they have an increased survival? Most of the formulations will just dissolve when added to water and therefore I do not expect a lasting effect of the encapsulation. Performing animal studies without a sound hypothesis of the effect to be expected seems almost unethical to me. 

It is unclear to me which encapsulated formulation is used throughout the animal and activity studies. 

Reviewer 2 Report

The paper submitted by Wang et al. deals with the study of effect of microencapsulation techniques on the stress resistance and biological activity of bovine lactoferricin-lactoferrampin-encoding Lactobacillus reuteri. The paper is clear, well written and the conclusions are supported by the results. However, some corrections are needed in order to increase the overall quality of the paper:

Comments on Figures, Figure Legends and Tables:

1.   Figure 5 legend: "L. reuteri" should be italicized. Also, whether bacterial colonization was detected in the intestines or in intestinal mucosa, please specify.

Comments on Methods:

1.   A method for analyzing the copy numbers of L. reuteri genes was not provided.

Comments on Results:

1.   LR-LFCA's resistance to harsh environments was detected for different lengths of time in different wall materials. Why is that?

Other specific comments:

1.   The authors didn't clearly describe how the vacuum environment affects bacterial activity in the discussion.

2.   The authors' presentation of the spray drying was still not adequate. The author should first comprehensively introduce the factors that affect the drying efficiency, and then the focus of this study should be introduced. The application of bovine lactoferrin peptide and Lactobacillus reuteri in actual production should be mentioned in the introduction.

3.   The protective effects of different wall materials (regardless of the final decision) on probiotics should be discussed and compared.
